# Effects of Radiation Intensity, Mineral Matrix, and Pre-Irradiation on the Bacterial Resistance to Gamma Irradiation under Low Temperature Conditions

**DOI:** 10.3390/microorganisms9010198

**Published:** 2021-01-19

**Authors:** Vladimir S. Cheptsov, Andrey A. Belov, Elena A. Vorobyova, Anatoli K. Pavlov, Vladimir N. Lomasov

**Affiliations:** 1Soil Science Faculty, Lomonosov Moscow State University, Leninskie Gory, 1, 12, 119234 Moscow, Russia; and.ant.be@gmail.com (A.A.B.); el.vb0247@gmail.com (E.A.V.); 2Space Research Institute, Russian Academy of Sciences, Profsoyuznaya str., 84/32, 117997 Moscow, Russia; 3Network of Researchers on the Chemical Evolution of Life, Leeds LS7 3RB, UK; 4Ioffe Physical-Technical Institute of the Russian Academy of Sciences, Polytechnicheskaya Street, 26, 194021 Saint-Petersburg, Russia; anatoli.pavlov@mail.ioffe.ru; 5STC “Nuclear Physics”, Peter the Great St. Petersburg State Polytechnic University, Polytechnicheskaya Street, 29, 195251 Saint-Petersburg, Russia; lomasoff@yandex.ru

**Keywords:** astrobiology, Mars, radioresistance, survivability, cryopreservation, ionizing radiation

## Abstract

Ionizing radiation is one of the main factors limiting the survival of microorganisms in extraterrestrial conditions. The survivability of microorganisms under irradiation depends significantly on the conditions, in which the irradiation occurs. In particular, temperature, pressure, oxygen and water concentrations are of great influence. However, the influence of factors such as the radiation intensity (in low-temperature conditions) and the type of mineral matrix, in which microorganisms are located, has been practically unstudied. It has been shown that the radioresistance of bacteria can increase after their exposure to sublethal doses and subsequent repair of damage under favorable conditions, however, such studies are also few and the influence of other factors of extraterrestrial space (temperature, pressure) was not studied in them. The viability of bacteria *Arthrobacter polychromogenes*, *Kocuria rosea* and *Xanthomonas* sp. after irradiation with gamma radiation at a dose of 1 kGy under conditions of low pressure (1 Torr) and low temperature (−50 °C) at different radiation intensities (4 vs. 0.8 kGy/h) with immobilization of bacteria on various mineral matrices (montmorillonite vs. analogue of lunar dust) has been studied. Native, previously non-irradiated strains, and strains that were previously irradiated with gamma radiation and subjected to 10 passages of cultivation on solid media were irradiated. The number of survived cells was determined by culturing on a solid medium. It has been shown that the radioresistance of bacteria depends significantly on the type of mineral matrix, on which they are immobilized, wherein montmorillonite contributes to an increased survivability in comparison with a silicate matrix. Survivability of the studied bacteria was found to increase with decreasing radiation intensity, despite the impossibility of active reparation processes under experimental conditions. Considering the low intensity of radiation on various space objects in comparison with radiobiological experiments, this suggests a longer preservation of the viable microorganisms outside the Earth than is commonly believed. An increase in bacterial radioresistance was revealed even after one cycle of irradiation of the strains and their subsequent cultivation under favourable conditions. This indicates the possibility of hypothetical microorganisms on Mars increasing their radioresistance.

## 1. Introduction

Habitability of extraterrestrial objects and the influence of factors of open space and regolith of cosmic bodies are the central issues of astrobiology [1,2,3,4]. Space missions carried out over the past decades have significantly expanded our understanding of the physicochemical conditions of outer space, regolith and atmospheres of planets and moons (in particular, Mars, Moon, Europa, Venus and others) [5,6]. To date, the cryogenic nature of many alien bodies and the limiting role of ionizing radiation in preservation and viability of microorganisms in extraterrestrial conditions have been shown [7,8]. The assessment of the radioresistance of microorganisms makes it possible to estimate the potential duration of their cryopreservation in a viable state on various space objects, and, therefore, to select promising space bodies and their regions, as well as rocks and depths for sampling during space missions in order to study the potential habitability of these space bodies [6,9]. The concept of radioresistance of terrestrial microorganisms also makes it possible to interpret the data obtained during space missions from the perspective of potential habitability of alien bodies in the past and to correct existing plans for colonization of extraterrestrial territories [10,11,12,13].

It was shown previously that resistance to ionizing radiation and the amount of radiation damage depend on the conditions of irradiation [14,15], which affect the amount and activity of free radicals, which are the main source (up to 90%) of radiation damage [16]. In particular, temperature affects the mobility of free radicals [14]. The content of water and oxygen influences free radicals abundance, since they are the main sources of free radicals [15,16]. Atmospheric pressure affects both the amount of О_2_ in the atmosphere and the amount of water in the sample by drying [15]. The conditions above are considered among the most important factors determining the severity of radiation damage to cells. It was also shown that the resistance of microorganisms in the native microbial communitiesin situin soils and rocks significantly exceeds that of pure (axenic) cultures to radiation [17]. It is assumed that this effect is caused by interactions between microorganisms, as well as between microorganisms, metabolites and the mineral matrix [15,17,18].

The influence of the characteristics of the mineral matrix on radioresistance of microorganisms, as well as on their resistance to the effects of a number of other factors of extraterrestrial space, has practically not been studied. It is likely that the mineralogical composition influences stability in several ways. Firstly, the adsorption (meaning adsorption capacity, pH, temperature, and salinity effects, types of bonding) of microorganisms on different minerals is different [19,20]. It is known that the physiology of microorganisms changes significantly in free and adsorbed states, and immobilization is capable of providing a protective effect [17,21,22]. Secondly, minerals can be a source of free radicals when exposed to ionizing radiation, and, therefore, affect the composition and number of radicals [23,24]. Thirdly, minerals, on the contrary, can trap free radicals [25]. Fourthly, the formation of toxins, or, on the contrary, substances useful for microorganisms in irradiated minerals could be assumed [26,27,28].

Previously, the influence of the type of mineral matrix on the effect of the open space conditions on microorganisms exposed in low Earth orbit (LEO) has been shown. In particular, the microcolonial black yeast-like fungus *Cryomyces antarcticus* (Selbmann et al., 2005) CCFEE 515 was able to withstand the combined stress of various extraterrestrial substrates, space, and simulated Mars-like conditions in terms of survival, DNA and ultrastructural stability. The proportion of viable cells of *C. antarcticus*, on average, was higher in the original substrate, from which this organism was isolated, and in the analogue of lunar regolith, in comparison with analogs of the modern and ancient Martian regoliths [29]. The effect of irradiation by helium and iron ions on the viability of *C. antarcticus* during immobilization on the same mineral carriers was studied [30,31]. It was found that the type of mineral matrix affects the indicators of cell viability (colony forming units number, integrity of cell membranes and DNA, enzyme activity), but what conditions provide the best survival of microorganisms remains unclear. The survival rates of *Deinococcus radiodurance* bacteria at immobilization in basalt, sandstone, forsterite, and fayalite were studied when exposed to ionizing radiation in a vacuum [32]. Under irradiation with 200 keV protons, the highest survival rate was observed in basalt and sandstone, and the lowest in fayalite. At irradiation with 4 keV carbon ions and 2 keV electrons, the effect of the mineral matrix on the bacteria survival was not detected.

Another poorly studied aspect of microorganisms’ radioresistance in space is the effect of radiation intensity. A significant limitation in many radiobiological experiments is their duration: in experiments, high doses of radiation accumulate over relatively short periods of time (from minutes to days), while in extraterrestrial conditions these doses accumulate over geological time (from thousands to millions of years) [14,33,34]. Obviously, it is impossible to overcome this limitation, which raises the question of influence of radiation intensity on microorganisms’ survival when the same absorbed radiation doses are accumulated. In experiments on irradiation under normal conditions, when metabolic processes are possible, it was shown that with a decrease in intensity, resistance increases due to the repair of damage during irradiation [14,35,36,37,38,39]. However, there are practically no studies on the effect of intensity of radiation exposure on the survival of microorganisms under conditions in which metabolic processes cannot proceed, in particular, at low ambient temperatures. In the study performed by Dartnell et al. [14], several radiation intensities were applied to gamma irradiation of several bacterial strains at a temperature of −79 °C, but this study did not set itself the goal of studying the effect of radiation intensity on the viability of microorganisms, and, therefore, the effect this factor has not been analyzed.

It is assumed that microorganisms, potentially existing on Mars, can exist in an ephemeral form, that is, they metabolize and reproduce only in rare periods of favorable conditions [17,37,40]. In this case, during the time elapsing between such favorable periods, they must accumulate significant radiation damage. It is known that radioresistance of microorganisms can be increased by the population accumulating multiple sublethal doses of radiation and subsequently reproducing in favorable conditions [41,42,43]. However, it should be noted that such studies are not numerous. They were only carried out for *Escherichia coli* (Migula 1895) and *Bacillus* spp., and did not consider the effects of any other astrobiological factors.

In this study, we irradiated strains of bacteria *Arthrobacter polychromogenes* (Schippers-Lammertse et al., 1963) SN_T61, *Kocuria rosea* (Flügge 1886) SN_T60 and *Xanthomonas* sp. DP3 by gamma radiation with a dose of 1 kGy under conditions of low pressure (1 Torr) and low temperature (−50 °C) at various radiation intensities (4 and 0.8 kGy/h) on various mineral matrices (montmorillonite and the analog of lunar dust). In the experiment, the native, previously non-irradiated strains and strains, previously irradiated with gamma radiation and subjected to 10 passages of culturing on solid media (as a model of short-term periods of favorable conditions between long periods of cryopreservation), were irradiated. The purpose of this study was to evaluate the effect of radiation intensity (4 vs. 0.8 kGy/h), mineral matrix (montmorillonite vs. analog of lunar dust) and pre-irradiation on the survival of the studied bacterial strains.

## 2. Materials and Methods

The objects of study were bacterial strains *A. polychromogenes* SN_T61, *K. rosea* SN_T60, and *Xanthomonas* sp. DP3. Strains *A. polychromogenes* SN_T61 and *K. rosea* SN_T60 were isolated from the arid soil of the Negev desert (Israel) [17], strain *Xanthomonas* sp. DP3 was isolated from soddy-podzolic soil (Moscow region, Russia) [44]. After isolation, the strains were maintained on glucose-peptone-yeast agar (GPY) [15].

Earlier, the strains *A. polychromogenes* SN_T61, *K. rosea* SN_T60 were immobilized in montmorillonite and irradiated with gamma radiation at a dose of 1 kGy at a radiation intensity of 0.5 kGy/h, a temperature of −50 °C, and a pressure of 1 Torr [45,46]. Further, the irradiated samples were cultured on GPY agar medium. Thus, pre-irradiated strains were obtained for the present study.

For the conduction of the experiment, the native strains (previously not irradiated) and pre-irradiated strains underwent 10 passages of culturing on GPY medium. The culturing was carried out at +28 °C for 3 days after each passage. The biomass was obtained after the 10th passage, suspended in sterile distilled water at a concentration of ~10^8^–10^9^ cells/mL. Then, 2 mL of the obtained suspension were introduced into 5 g of montmorillonite or an analog of lunar dust in polypropylene containers, the samples with suspension were thoroughly mixed with sterile glass rods, dried at +28 °C for three days to obtain air-dry samples, and then divided into the subsamples.

Montmorillonite is a clay aluminosilicate mineral found on Mars and the regions of its discovery are very promising from an astrobiological point of view [47,48,49,50]. The mineral for the experiment was obtained from Nanoshel LLC (Wilmington, DE, USA) and contained >99% of Al_2_[(OH)_2_Si_4_O_10_]·nH_2_O. The analog of lunar dust is represented by microspheres of silicon oxide up to 50 μm in size [51] and is similar to the simulators of lunar dust previously used in a number of other studies [52]. It was supplied by American Elements (Los Angeles, CA, USA) and contained >99.9% of SiO_2_. Before the introduction of the microorganisms, the mineral matrices were sterilized by calcining at +600 °C for 3 h.

For irradiation, the samples of 200 mg in mass in triplicates were placed into a previously described climatic chamber [53], which allows irradiation with gamma radiation in low pressure and low temperature conditions. The climatic chamber is a forevacuum chamber with a zeolite cryogenic pump inside for effective capturing of the water vapour and other gases. The forevacuum chamber is surrounded by a jacket filled with liquid nitrogen. The chamber contains a cylinder about 1 cm in diameter and about 12 cm long divided on two similar parts, in which samples can be placed simultaneously. Irradiation was performed on a K-120000 gamma facility with ^60^Co sources. Two irradiation trials were performed: the first one at radiation intensity of 0.8 kGy/h and the second one at 4 kGy/h. The temperature of the samples under irradiation was −50 °C, the pressure in the chamber was 1 Torr. The pressure and temperature conditions were identical at both irradiation trials. The samples were irradiated with a dose of 1 kGy. Non-irradiated samples exposed under the same pressure and temperature for 1 h as well as the samples not subjected to the extreme factors above served as controls.

The number of viable bacteria was estimated by direct plating of sample suspensions onto GPY agar. Before inoculation, the microorganisms were desorbed from mineral particles by processing the sample suspensions on a Heidolph Multi Reax vortex for 30 min at 2000 rpm, as was described previously [17,34]. Such desorption regime was found optimal in course of previous studies. Suspensions of samples in different dilutions were plated in triplicates with simultaneous control of the nutrient medium sterility, sterility of the water used for dilutions preparation, and control of the presence of foreign air microflora. The culturing was carried out at +28 °C for 14 days.

Statistical processing of the data was carried out using Microsoft Office Excel 2007. For each treatment option, the mean number of colony forming units per gram (CFU/g) and the standard deviation of the mean (*p* < 0.05) were calculated. A Student’s *t*-test was used to compare data. Due to the fact that the number of colony-forming units in the control samples of different strains and on different matrices was different, for convenience they were also expressed as a decrease in the number (logarithmic reduction) relative to the control values (Figure 1) and the percentage of microorganisms survived (Table A1). In these calculations, the CFU numbers in the samples exposed under low pressure and low temperature without irradiation were taken as control values.

## 3. Results

Exposure under low pressure and low temperature conditions without irradiation did not affect the number of viable microorganisms—the CFU numbers were within 98–101% of that in unexposed samples.

In general, all the studied strains demonstrated high resistance and retained a high number of living cells under all irradiation treatments. *Arthrobacter polychromogenes* and *K. rosea* showed the highest resistance to irradiation under both irradiation intensities and on both mineral carriers retaining 13–93% and 8–80% of the number of viable cells from the control, respectively, depending on the conditions of irradiation and pre-irradiation. *Xantomonas* sp. showed a significantly higher sensitivity to radiation: the decline in numbers reached almost five orders of magnitude under some exposure regimes. The highest resistance of *Xantomonas* sp. was observed at immobilization on montmorillonite at a radiation intensity of 0.8 kGy/h—under these conditions, the decrease in CFU number was about one order of magnitude (Figure 1b and Table A1). After irradiation of *A. polychromogenes* SN_T61 and *K. rosea* SN_T60 strains immobilized on both investigated mineral carriers, at both radiation intensities, a 1.2–2.9 times higher decrease in the number of viable cells of variants that had not been pre-irradiated, was observed (Figure 1a). Moreover, the effect of pre-irradiation was more pronounced during immobilization in montmorillonite: the proportion of surviving cells of pre-irradiated strains was 1.6–2.9 times higher than the proportion of surviving cells of non-irradiated strains, while during immobilization in the analogue of lunar dust these indicators differed by 1.2–1.6 times.

For all strains on both mineral matrices, a higher proportion of living cells remained at a lower radiation intensity. For *A. polychromogenes*, the proportion of cells that survived after irradiation at 0.8 kGy/h was 1.1–2.8 times higher than that at 4 kGy/h; for *K. rosea*, similar indicators differed by 2.1–3 times. The most pronounced effect of radiation intensity was observed for *Xanthomonas* sp.—at a lower radiation intensity, 36–128 times more cells survived (Figure 1 and Table A1).

The dependence of bacterial radioresistance on the type of mineral matrix was also the same for all strains: the number of viable cells after irradiation in the variants immobilized on montmorillonite was higher than in the variants immobilized on the analog of lunar dust. For *A. polychromogenes* and *K. rosea*, the number of CFU under irradiation in montmorillonite exceeded that under irradiation in silicon oxide by 1.2–3.8 times. In most cases, these differences were better expressed in the pre-irradiated strains—the number on different carriers differed by 2.2–3.8 times, for non-irradiated strains—by 1.2–3 times. The most dramatic differences in resistance were found for *Xanthomonas* sp.—in montmorillonite, 59 and 210 times more cells were preserved than in the analogue of lunar dust, after irradiation at intensities of 0.8 kGy/h and 4 kGy/h, respectively.

## 4. Discussion

The observed complete preservation of CFU numbers of bacteria after exposure to low pressure and low temperature is similar with the previous results on the absence of a significant effect of 0.01 Torr pressure and −130 °C temperature on the viability of bacteria immobilized in montmorillonite [17].

The studied strains retained a high number of viable cells after irradiation with a dose of 1 kGy. This is well agreed with the previously obtained data on the high resistance of pure cultures of soil bacteria (including strains *A. polychromogenes* SN_T61, *K. rosea* SN_T60) to irradiation with doses of up to 10 kGy under similar conditions [17,45,46,54].

*Arthrobacter polychromogenes* and *K. rosea* demonstrated higher resistance than *Xanthomonas* sp. *Kocuria rosea* and *A. polychromogenes* belong to the phylum *Actinobacteria*, representatives of which are known for their high abundance in various extreme habitats and resistance to various stress effects [55,56,57,58,59], which explains their higher resistance to radiation, taking into account the relationship of various mechanisms of stress-tolerance with radioresistance [37,60,61,62]. Moreover, these strains were isolated from desert soil, in contrast to *Xanthomonas* sp., isolated from a non-extreme ecotope, and microorganisms from extreme habitats may have more developed mechanisms of resistance [63].

Previously it was shown that the radioresistance increase of *E. coli* and *Bacillus* spp. occurs only after several (3 and more) cycles of irradiation and outgrowth and it was suggested that acquisition of increased resistance phenotype requires several genomic alterations [41,42,43]. In our case, radioresistance increase was observed even after one cycle of pre-irradiation. It should also be noted that in the studies above the strains were pre-irradiated with sublethal doses killing ~99.9% of populations. In our experiments no decrease in the number of viable cells of *A. polychromogenes* SN_T61 and *K. rosea* SN_T60 was detected during pre-irradiation with 1 kGy dose [45,46] and it was found that these strains are able to withstand the effect of an order of magnitude higher dose of ionizing radiation [17]. Thus, an increase in radioresistance is possible even after one cycle of pre-irradiation and don’t necessarily require accumulation of the sublethal doses of radiation.

As mentioned in the introduction, there are suggestions that the existence of microorganisms on Mars is ephemeral: presented by sequential alternation of periods of metabolic rest and activity caused by changes in the external physicochemical environment [41]. With such model of existence, accumulation of high doses of radiation and, as a consequence, the accumulation of radiation damage could exist during periods of metabolic dormancy, with its subsequent reparation and an increase in radioresistance, during periods of metabolic activity. Microorganisms’ radiotolerance increase in response to exposure to high doses of ionizing radiation and its persistence in time testifies in favor of the potential habitability of Mars [40,64]. If there were terrestrial-like microorganisms in the previous geological periods, then during the gradual processes of the planet’s atmosphere and magnetic field degradation, followed by the increasing level of cosmic radiation [65,66,67], bacteria could acquire radioresistant properties. If this is the case, one can assume the presence of these communities in the regolith in a preserved or metabolically active form till now, as well as an increase in their radioresistance at the present time. At the same time, apparently, the presence of even a very small number of short periods favorable for the growth of microorganisms can contribute to a significant increase in resistance.

For all the strains studied in the both mineral carriers, there is a general tendency of increasing survival rate with decreasing radiation intensity. This pattern has been previously shown during irradiation under conditions under which metabolic processes are possible, and the increase in radioresistance was explained by the repair of damage during irradiation [14,35,36,37,38,39]. However, it is unlikely that a similar process was possible in our experiment. There is quite a lot of information on the activity of various metabolic processes in microbial cells at temperatures as low as −40 °C [6,68,69,70], and in this regard, the implementation of any metabolic processes at −50 °C (the temperature in our experiment), in general, seems possible. But even if at such a temperature the damage repair occurs, it must be extremely slow, given the known data on low rate of metabolic processes at low temperatures [69,70,71] and probably does not significantly contribute to the change in survival, taking into account the duration of the radiation dose accumulation (0.25 h and 1.25 h for 4 kGy/h and 0.8 kGy/h, respectively). In addition to low temperatures, possible metabolic activity could also be inhibited due to low pressure. Thus, it is likely that the change in the survival rate of microorganisms with a change in the irradiation intensity under low-temperature conditions is associated with the changes in the kinetics of free radicals’ formation, their recombination and reactions with biomolecules, and in the kinetics of interactions between the products of these reactions. Obviously, the observed effect requires additional research, both in terms of analysing its presence at other temperatures, different types of radiation, for other organisms, etc., and from the point of view of studying the dependences of the reaction kinetics on the radiation intensity under low-temperature conditions.

The intensity of ionizing radiation on Mars, Europa, meteorites, is, usually, from few mGy to few Gy per year [15,72,73,74], which is orders of magnitude lower than the radiation intensities used in radiobiological experiments ([14,34] and references therein). In this regard, it can be assumed that under these conditions the observed effect could be much more pronounced, and the duration of the survival of microorganisms in the composition of cosmic bodies in a cryopreserved state can significantly exceed current estimates [14,15,17,44,72].

A decrease in the number of viable cells depending on the type of the mineral matrix was a general tendency for all studied strains: the number of viable cells in the variants immobilized on montmorillonite was higher than in the variants immobilized on the analogue of lunar dust. As indicated above, there are numerous ways by which the mineral matrix can affect microbial survivability under irradiation, and it was not part of the study to identify the mechanisms of mineral matrix action. However, it was shown that this effect can be significant: for *Xanthomonas* sp. in montmorillonite, 59–210 more living cells survived compared to variants with the silicate matrix. Consequently, to determine the most promising regions and types of deposits for the search for life on Mars (or other extraterrestrial bodies), it is necessary to study the stability of microorganisms on some mineral carriers found on the planets or moons. The results of our study indicate that such promising sediments could be the clay minerals, especially taking into account their formation under conditions favorable for microorganisms [74,75], their ability to adsorb water vapour from the atmosphere [49] and the possibility of them being used by microorganisms a source of nutrients [47].

To summarize, our study showed that the radioresistance of bacteria under low pressure and low temperature conditions significantly depends on the type of mineral matrix, on which they are immobilized, while montmorillonite promotes increased survival in comparison with a silicate matrix. Survival depends on the intensity of radiation, despite the impossibility of active reparation processes. Given the low intensity of radiation on various space objects in comparison with radiobiological experiments, there is a reason to assume a longer preservation of microorganisms’ viability than is commonly believed. The reasons for this effect are currently unclear and require additional experiments. An increase in bacterial radioresistance was revealed even after one cycle of strains irradiation under simulated Martian conditions and culturing under favorable conditions. This indicates the possibility of increasing radioresistance by hypothetical microorganisms on Mars.

## Figures and Tables

**Figure 1 microorganisms-09-00198-f001:**
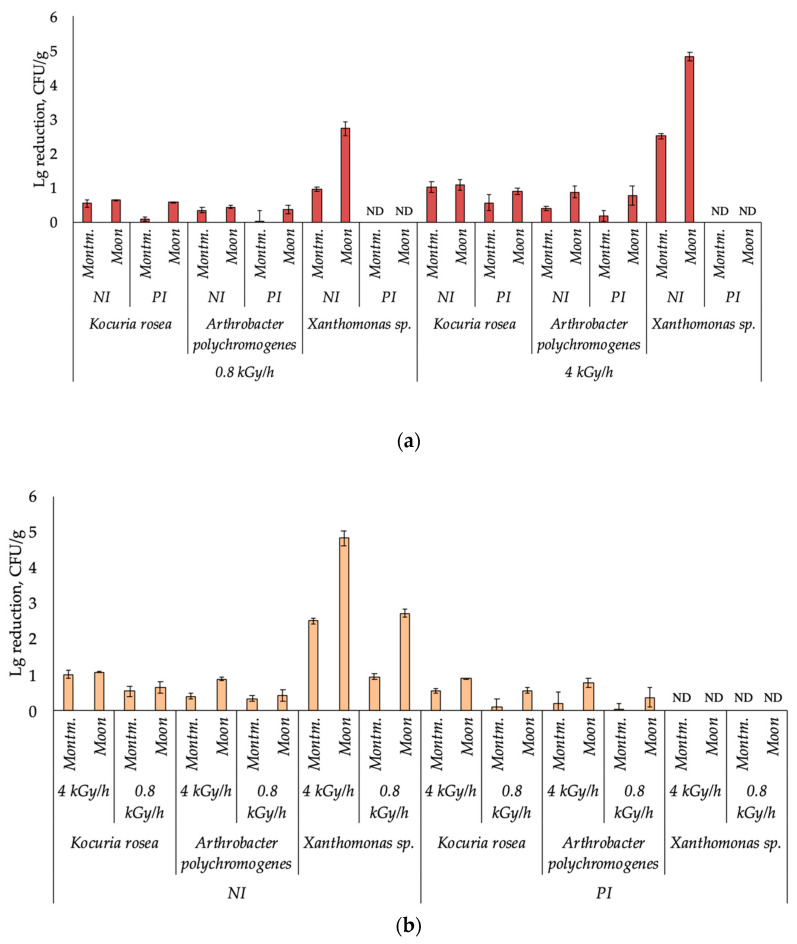
Decrease in the number (logarithmic reduction) of viable bacteria under different conditions of irradiation (subfigures (**a**,**b**) are based on the same data, combined in different ways for comparison convenience). Montm.—variants immobilized on montmorillonite; Moon—variants immobilized on lunar dust analog; NI—non-pre-irradiated strains; PI—pre-irradiated strains. ND—no data.

## Data Availability

Data sharing not applicable.

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
