# Peer review of "Effects of Radiation Intensity, Mineral Matrix, and Pre-Irradiation on the Bacterial Resistance to Gamma Irradiation under Low Temperature Conditions"

_microorganisms, 2021, doi:10.3390/microorganisms9010198_

Round 1

Reviewer 1 Report

The manuscript entitled “Effects of Radiation Intensity, Mineral Matrix, and Pre-irradiation on the Bacterial Resistance to Gamma Irradiation Under Low-Temperature Conditions” is well written, easy to follow, and clearly presented. It brings novel data to the discussion about the radiation resistance of microorganisms in more realistic scenarios using biological models isolated from natural extreme environments. This is highly relevant for understanding the resilience of life in general. More specifically, it represents a contribution to the discussion about planetary protection, the habitability of extraterrestrial environments, and somewhat relates to other topics such as, in situ resource utilization (ISRU), biomining, etc.

I have only a few minor comments listed below:

1) Line 69: Precisely because the literature on this particular topic is so scarce, I think it would be worth noting data available in the literature about the effects of different matrices on the survival of microorganisms against radiation. For example, Paulino-Lima et al. (2011) reported on the effects of basalt, sandstone, forsterite (Mg2SiO4), and fayalite (Fe2SiO4) on the survival of Deinococcus radiodurans against ionizing radiation (charged particles) under low-pressure conditions (Paulino-Lima et al. (2011) Survival of Deinococcus radiodurans against Laboratory-Simulated Solar Wind Charged Particles, https://doi.org/10.1089/ast.2011.0649)

2) Line 156: The process of desorption from mineral particles was performed by processing the sample suspensions on a Heidolph Multi Reax vortex for 30 min at 2000 rpm. The authors cite reference #17 for this process. I wonder if any optimization was made, considering intermediate steps from zero to 30 minutes? How would this process affect the viability of the cells in control samples exposed to all conditions except irradiation?

3) Line 161: CFU must be defined at its first occurrence.

Author Response

Dear Reviewer,

Thank you for your comments and suggestions! We have taken them into account and performed appropriate corrections. We hope the MS is acceptable now; otherwise we are ready to continue improvement of the MS. Please find detailed list of corrections below. For the convenience your comments are marked like “Q” and our corrections/answers are marked like “A”. Since the line numbering in the revised MS may change depending on the version of Microsoft Word or other software used for the file viewing, we give the line numbering according to the PDF file attached.

Q: The manuscript entitled “Effects of Radiation Intensity, Mineral Matrix, and Pre-irradiation on the Bacterial Resistance to Gamma Irradiation Under Low-Temperature Conditions” is well written, easy to follow, and clearly presented. It brings novel data to the discussion about the radiation resistance of microorganisms in more realistic scenarios using biological models isolated from natural extreme environments. This is highly relevant for understanding the resilience of life in general. More specifically, it represents a contribution to the discussion about planetary protection, the habitability of extraterrestrial environments, and somewhat relates to other topics such as, in situ resource utilization (ISRU), biomining, etc.

A: Thank you!

Q: I have only a few minor comments listed below:

 1) Line 69: Precisely because the literature on this particular topic is so scarce, I think it would be worth noting data available in the literature about the effects of different matrices on the survival of microorganisms against radiation. For example, Paulino-Lima et al. (2011) reported on the effects of basalt, sandstone, forsterite (Mg2SiO4), and fayalite (Fe2SiO4) on the survival of Deinococcus radiodurans against ionizing radiation (charged particles) under low-pressure conditions (Paulino-Lima et al. (2011) Survival of Deinococcus radiodurans against Laboratory-Simulated Solar Wind Charged Particles, https://doi.org/10.1089/ast.2011.0649)

A: Thank you! We have added these data in lines 91-96:

“The survival rates of Deinococcus radiodurance bacteria at immobilization in basalt, sandstone, forsterite, and fayalite were studied when exposed to ionizing radiation in a vacuum [32]. Under irradiation with 200 keV protons, the highest survival rate was observed in basalt and sandstone, and the lowest in fayalite. At irradiation with 4 keV carbon ions and 2 keV electrons, the effect of the mineral matrix on the bacteria survival was not detected.”

Q: 2) Line 156: The process of desorption from mineral particles was performed by processing the sample suspensions on a Heidolph Multi Reax vortex for 30 min at 2000 rpm. The authors cite reference #17 for this process. I wonder if any optimization was made, considering intermediate steps from zero to 30 minutes? How would this process affect the viability of the cells in control samples exposed to all conditions except irradiation?

A: The optimization of the desorption technique was performed in course of previous studies of soil microbial communities and immobilized bacterial strains. To date there are two the most commonly used ways for cells desorption in soil microbiology: ultrasonication and vortexing. But ultrasonication provides higher risk of the sample contamination, so we use vortexing through several last years. The trials with different times and rpm were performed to find the regime leading to the highest CFU numbers (for soils and immobilized pure cultures) and the highest morphological biodiversity (for soils). The vortexing for 30 min at 2000 rpm was found optimal and comparable in results with standard ultrasonication procedures. Some comments are added in lines 206-207, as well as the reference for paper containing discussion of the issue is added: “Before inoculation, the microorganisms were desorbed from mineral particles by processing the sample suspensions on a Heidolph Multi Reax vortex for 30 min at 2000 rpm, as was described previously [17,34]. Such desorption regime was found optimal in course of previous studies.”

Regarding intermediate steps from zero to 30 min, the fewer times could lead to incomplete desorption of the cells and, consequently, less number of CFUs observed.

Q: 3) Line 161: CFU must be defined at its first occurrence.

A: CFU is defined now in lines 89-90 and 212.

Reviewer 2 Report

The present written by Cheptsov et al. studiess the effects of gamma radiation on the survivability of three bacterial strains under conditions of low temperature and low pressure. The authors investigated how different radiation intensities affect the survival of microorganisms on two different mineral surfaces. Moreover, they also studied how previously gamma irradiated bacterial strains responds to different intensity radiation on mineral surfaces. 

Most of these effects are already reported in previous literatures. The effect of temperature and pressure on radioresistance of bacteria is already known (14, 15). Preirradiation is known to increase the radioresistance of other microorganisms (40-46). Immobilization from adsorption e.g., in soils and rocks increases the resistance of microorganism to radiation (17, 21, 22). It is also known that different mineral types affect radioresistance of microorganism (30, 31). Thus, the results presented here are quite predictable. My other concerns are listed below:

  1. It is hard to predict from the experimental scheme whether dependence of resistance on radiation intensity is due to temperature, or pressure or due to the mineral effect? More systematic series of control experiments like experiments in ambient temperature and pressure in presence or absence of mineral is required.

  1. The authors should show standard deviations for each CFU values in table A1.

  1. Method of immobilization on mineral should be clearly described in the method section.

  1. The authors immobilize the bacteria on mineral during pre-irradiation however used mineral suspension with the biomass for subsequent irradiation. The reason for this difference needs to be clearly stated. It is not clear whether the effect is due to mineral contact or from adsorbed microorganisms?

Author Response

Dear Reviewer,

Thank you for your comments and suggestions! We have taken them into account and performed appropriate corrections. We hope the MS is acceptable now; otherwise we are ready to continue improvement of the MS. Please find detailed list of corrections below. For the convenience your comments are marked like “Q” and our corrections/answers are marked like “A”. Since the line numbering in the revised MS may change depending on the version of Microsoft Word or other software used for the file viewing, we give the line numbering according to the PDF file attached.

Q: The present written by Cheptsov et al. studiess the effects of gamma radiation on the survivability of three bacterial strains under conditions of low temperature and low pressure. The authors investigated how different radiation intensities affect the survival of microorganisms on two different mineral surfaces. Moreover, they also studied how previously gamma irradiated bacterial strains responds to different intensity radiation on mineral surfaces. 

Most of these effects are already reported in previous literatures. The effect of temperature and pressure on radioresistance of bacteria is already known (14, 15). Preirradiation is known to increase the radioresistance of other microorganisms (40-46). Immobilization from adsorption e.g., in soils and rocks increases the resistance of microorganism to radiation (17, 21, 22). It is also known that different mineral types affect radioresistance of microorganism (30, 31). Thus, the results presented here are quite predictable. My other concerns are listed below:

A: It is known that temperature and pressure affect bacterial radioresistance, but there are only few studies simultaneously reproducing effects of low temperature, low pressure, and high-dose irradiation. In fact, all these studies are performed by us (Cheptsov et al., 2017; 2018a; 2018b; 2018c; Pavlov et al., 2019). These studies shown increase in radiotolerance under such irradiation conditions (comparing with ambient conditions). Nevertheless, the limits of radioresistance under such conditions, changes in resistance of wide diversity of microorganisms, effect of a number of other factors in combination with factors above are unstudied. It was shown, that some physical factors can lead to synergetic effect, so further studies (including the present studies) are needed to clarify the effects of a lot of astrobiologically relevant factors and to improve astrobiological models. Pre-irradiation experiments were earlier performed only for Bacillus spp. and Escherichia coli under normal conditions with dozens cycles of irradiation and reparation – but are other diverse bacteria able to increase their radioresistance in the same manner when they are irradiated under Mars-like conditions? How rapidly their radioresistance could increase? Does the radioresistance increasing under such conditions require a lot of irradiation-reparation cycles (considering hypotheses about past Martian climate it could require millions years, during which the cells could dead due to irradiation) or radioresistanse could increase even at sparse presence of periods of microbial metabolic activity (short discussion is added on lines 285-294)? Data on the mineral matrix effect on the microbial radioresistance are very scarce, and there are no clear patterns found (see lines 80-96, please). So it is obvious that studies are needed to find the minerals which provide increased radioresistance and, consequently, to define the rocks/regions on Mars where living cells could be found (it is discussed in lines 395-400). Effects of irradiation intensity on microbial survivability under low-temperature conditions was not studied previously, and the effect found can significantly affect the current estimations of possible duration of microorganisms’ viable cryopreservation under extraterrestrial conditions. In our view (and in accordance with Reviewer 1 and Reviewer 3) such studies are necessary to improve astrobiological scenarios and the presented study has novelty and contributes significantly to the current knowledge on the microbial resistance under extraterrestrial conditions. The most of the research in the world is devoted to clarifying previously discovered patterns, a detailed study of any known effects, solving regional problems, etc. - it is "normal science". If it is assumed that after the discovery of some general effect the further research should not be carried out, then in this way it is possible to discard the majority of published scientific articles and lose huge amounts of scientific knowledge. We hope that you will reconsider your opinion about the paper novelty and significance of the results.

Q: 1. It is hard to predict from the experimental scheme whether dependence of resistance on radiation intensity is due to temperature, or pressure or due to the mineral effect? More systematic series of control experiments like experiments in ambient temperature and pressure in presence or absence of mineral is required.

A: The temperature and pressure conditions during irradiation were identical for both radiation intensities. The similar dependence of resistance on radiation intensity was observed at immobilization on both mineral carriers. To highlight it we have modified the sentences on lines 197-201: “Irradiation was performed on a K-120000 gamma facility with 60Co sources. Two irradiation trials were performed: the first one at radiation intensity of 0.8 kGy/h and the second one at 4 kGy/h. The temperature of the samples under irradiation was -50 °C, the pressure in the chamber was 1 Torr. The pressure and temperature conditions were identical at both irradiation trials.” and on lines 311-312: “For all the strains studied in the both mineral carriers, there is a general tendency of increasing survival rate with decreasing radiation intensity.”

Q: 2.The authors should show standard deviations for each CFU values in table A1.

A: The standard deviations are added to the Table A1 (line 440).

Q: 3. Method of immobilization on mineral should be clearly described in the method section.

A: We have added some details regarding immobilization on lines 160-163: “Then, 2 mL of the obtained suspension were introduced into 5 g of montmorillonite or an analog of lunar dust in polypropylene containers, the samples with suspension were thoroughly mixed with sterile glass rods, dried at +28 °C for three days to obtain air-dry samples, and then divided into the subsamples.”. In our view all the possible details are indicated now. If there is still any unclearness, please specify which parameters and/or operations are unclear – we will clarify it.

Q: 4.The authors immobilize the bacteria on mineral during pre-irradiation however used mineral suspension with the biomass for subsequent irradiation. The reason for this difference needs to be clearly stated. It is not clear whether the effect is due to mineral contact or from adsorbed microorganisms?

A: The immobilization was performed identically, and in all cases the dried samples were irradiated, as it was indicated on line 141. So there was no difference. To highlight it we have modified the sentence on lines 162-163: “…dried at +28 °C for three days to obtain air-dry samples…”.

References:

Cheptsov, V., Vorobyova, E., Belov, A., Pavlov, A., Tsurkov, D., Lomasov, V., & Bulat, S. (2018a). Survivability of soil and permafrost microbial communities after irradiation with accelerated electrons under simulated Martian and open space conditions. Geosciences, 8(8), 298.

Cheptsov, V. S., Vorobyova, E. A., Gorlenko, M. V., Manucharova, N. A., Pavlov, A. K., & Lomasov, V. N. (2018b). Effect of gamma radiation on viability of a soil microbial community under conditions of Mars. Paleontological Journal, 52(10), 1217-1223.

Cheptsov, V. S., Vorobyova, E. A., Osipov, G. A., Manucharova, N. A., Polyanskaya, L. M., Gorlenko, M. V., ... & Lomasov, V. N. (2018c). Microbial activity in Martian analog soils after ionizing radiation: Implications for the preservation of subsurface life on Mars. AIMS microbiology, 4(3), 541.

Cheptsov, V. S., Vorobyova, E. A., Manucharova, N. A., Gorlenko, M. V., Pavlov, A. K., Vdovina, M. A., ... & Bulat, S. A. (2017). 100 kGy gamma-affected microbial communities within the ancient Arctic permafrost under simulated Martian conditions. Extremophiles, 21(6), 1057-1067.

Pavlov, A., Cheptsov, V., Tsurkov, D., Lomasov, V., Frolov, D., & Vasiliev, G. (2019). Survival of Radioresistant Bacteria on Europa’s Surface after Pulse Ejection of Subsurface Ocean Water. Geosciences, 9(1), 9.

Reviewer 3 Report

Dear Authors,

This is well designed study and well written paper. I have only a minor comments and corrections. All of them are marked as a track changes in the attached file.

Author Response

Dear Reviewer,

Thank you for your comments and suggestions! We have taken them into account and performed appropriate corrections. We hope the MS is acceptable now; otherwise we are ready to continue improvement of the MS. Please find detailed list of corrections below. For the convenience your comments are marked like “Q” and our corrections/answers are marked like “A”. Since the line numbering in the revised MS may change depending on the version of Microsoft Word or other software used for the file viewing, we give the line numbering according to the PDF file attached.

Q: This is well designed study and well written paper. I have only a minor comments and corrections. All of them are marked as a track changes in the attached file.

A: Thank you!

Q: Line 58 - irradiation by

A: Line 58 – Corrected as suggested.

Q: Lines 60-65 - This sentence is long and hard to understand. Please re-write it.

A: The sentence is divided into several sentences (Lines 60-65): “In particular, temperature affects the mobility of free radicals [14]. The content of water and oxygen influences free radicals abundance, since they are the main sources of free radicals [15,16]. Atmospheric pressure affects both the amount of О2 in the atmosphere and the amount of water in the sample by drying [15]. The conditions above are considered among the most important factors determining the severity of radiation damage to cells.”

Q: Line 82 - I suggest to give the author and year who described this species when you use it for the first time in the text.

A: Corrected as suggested (line 82).

Q: Lines 92-93 - (experiments on irradiation)

A: Lines 114 – Corrected as suggested

Q: Line 113 - author/s and year

A: Corrected as suggested (line 134).

Q: Line 115 – as above

A: Corrected as suggested (line 137).

Q: Lines 118-124 - In my opinion this part is not necessary in the Introduction. This is rather part of M&M section.

A: In our view it would be better to keep this part in Introduction, since it explains the aims of the study to the readers and makes the following M&M clearer. If this part will be removed from Introduction, the Introduction will not describe one of the aims – study of preirradiation effect. At the same time, the explanation of this goal requires brief scheme of the experiment. So we would prefer to keep it here, but if it is essential, we are ready to move this part to M&M section.

Q: Lines 126, 131 – Arthrobacter, Kocuria

A: Corrected as suggested (lines 147, 152).

Q: Line 140 - Please specify how you prepared such dust or where you bought it. Please also give a physical and chemical characteristic of this dust.

A: We have added it on lines 184-186: “The mineral for the experiment was obtained from Nanoshel LLC (Wilmington, DE, USA) and contained >99% of Al2[(OH)2Si4O10]·nH2O.”

Q: Line 140 - Please specify how you prepared such dust or where you bought it. Please also give a physical and chemical characteristic of this dust.

A: We have added it on lines 188-189: “It was supplied by American Elements (Los Angeles, CA, USA) and contained >99.9% of SiO2.”

Q: Line 149 - Please explain at least in short how this chamber looks like and works etc..

A: The chamber was described in detail previously, and the reference for its detailed description is given. At the same time we agree that a brief description of the chamber will clarify how it works. Such description is added on lines 193-197: “The climatic chamber is a forevacuum chamber with a zeolite cryogenic pump inside for effective capturing of the water vapour and other gases. The forevacuum chamber is surrounded by a jacket filled with liquid nitrogen. The chamber contains a cylinder about 1 cm in diameter and about 12 cm long divided on two similar parts, in which samples can be placed simultaneously.”

Q: Line 153 - Are these samples were also closed in the chamber? Please explain this in more details in the text.

A: Special thanks for this comment! The samples exposed in the chamber without irradiation as well as the samples not exposed in the chamber served as controls. We also have performed controls in the same way previously in other experiments and no significant effect of similar exposure was observed (Cheptsov et al., 2018 – reference #17 in the MS). In the present study the differences between two types of controls were negligible (they were up to 2% only). Due to it the log-reductions and the percentages of microorganisms survived relatively to the chamber-exposed control only are shown to avoid complication of the figures and table. However, this should definitely be indicated in the manuscript and explained. We have overlooked absence of its explanation at the MS preparation since the data on the both types of controls were in fact identical - thanks for paying attention on it. Appropriate corrections are performed:

lines 201-203: “Non-irradiated samples exposed under the same pressure and temperature for 1 h as well as the samples not subjected to the extreme factors above served as controls.”,

lines 217-218: “In these calculations, the CFU numbers in the samples exposed under low pressure and low temperature without irradiation were taken as control values.”,

lines 220-222: “Exposure under low pressure and low temperature conditions without irradiation did not affect the number of viable microorganisms - the CFU numbers were within 98-101% of that in unexposed samples (data not shown).”,

lines 269-272: “The observed complete preservation of CFU numbers of bacteria after exposure to low pressure and low temperature is similar with the previous results on the absence of a significant effect of 0.01 Torr pressure and -130°C temperature on the viability of bacteria immobilized in montmorillonite [17].”.

Q: Lines 160-165 - Did you used any specific statistical tests? Please explain which one.

A: We have added it on line 213: “A Student’s t-test was used to compare data.”

Q: Line 168 - When you start new sentence the name of the genus need to be in full. You can not start the sentence with abbreviation.

A: Corrected as suggested (line 224).

Q: Line 171 – greater  higher

A: Corrected as suggested (line 227).

Q: Line 173 – options regimes

A: Corrected as suggested (line 229).

Q: Line 177 – greater  higher

A: Corrected as suggested (line 233).

Q: Line 197 (Figure 1) - I suggest to use a full genus names.

A: Corrected as suggested (Line 266).

Q: Line 198 - In my opinion this section need to be re-written because many fragments are just repetitions from Introduction and authors do not discuss their own results but results of other studies. Moreover, this section is also full of speculations without any proofs from present study. I marked some of them below.

A: The section is corrected – please refer to the detailed comments below.

Q: Lines 199-201 - This study examined the effect of the mineral matrix, the intensity of radiation and the effect of pre-irradiation on the viability of pure cultures of bacteria under irradiation under conditions that simulate the Martian regolith.

A: Corrected as suggested – the sentence is deleted.

Q: Line 205 - See comment above on the abbreviations on the sentences beginning.

A: Corrected – the full genus name is written (line 277).

Q: Line 205 - as above

A: Corrected – the full genus name is written (line 278).

Q: Line 208 – greater  higher

A: Corrected as suggested (line 280).

Q: Line 210 – while and

A: Corrected as suggested (line 283).

Q: Lines 214-215 - This is a pure speculation without any proofs in present study.

A: This part is completely rewritten – lines 285-294:Previously it was shown that the radioresistance increase of E. coli and Bacillus spp. occurs only after several (3 and more) cycles of irradiation and outgrowth and it was suggested that acquisition of increased resistance phenotype requires several genomic alterations [41,42,43]. In our case, radioresistance increase was observed even after one cycle of pre-irradiation. It should also be noted that in the studies above the strains were pre-irradiated with sublethal doses killing ~99.9% of populations. In our experiments no decrease in the number of viable cells of A. polychromogenes SN_T61 and K. rosea SN_T60 was detected during pre-irradiation with 1 kGy dose [45,46] and it was found that these strains are able to withstand the effect of an order of magnitude higher dose of ionizing radiation [17]. Thus, an increase in radioresistance is possible even after one cycle of pre-irradiation and don't necessarily require accumulation of the sub-lethal doses of radiation”.

Q: Lines 217-220 - one again a pure speculation

A: This part is completely rewritten – lines 285-294 (please refer to the comment above).

Q: Line 223 – mode model

A: Corrected as suggested (line 298).

Q: Line 226 – Authors – the text is highlighted, but a comment is absent

A: We suggest that you have meant that the sentence was too long and hard to read. We rephrased it and divide in two sentences: “Microorganisms’ radiotolerance increase in response to exposure to high doses of ionizing radiation and its persistence in time testifies in favor of the potential habitability of Mars [40,64]. If there were terrestrial-like microorganisms in the previous geological periods, then during the gradual processes of the planet's atmosphere and magnetic field degradation, followed by the increasing level of cosmic radiation [65-67], bacteria could acquire radioresistant properties.” – Lines 301-306.

Q: Line 236 - in this work

A: Corrected as suggested (line 311).

Q: Line 240 – about on

A: Corrected as suggested (line 315).

Q: Line 256 - in space conditions, including

A: Corrected as suggested (line 382).

Q: Line 271 – add “planets or moons”

A: Corrected as suggested (line 397).

Q: Line 274 - as sources a source

A: Corrected as suggested (line 400).

Round 2

Reviewer 2 Report

The authors have addressed my concern. I recommend this paper for publication.